# KEY POINT IS KEY IN RESOLVING THE OFFLINE THREE-DIMENSIONAL BIN PACKING PROBLEM

## ABSTRACT

In recent years, with the surge in deep learning and reinforcement learning, researchers have begun to explore the use of deep reinforcement learning to solve the offline three-dimensional bin packing problem. However, the valid action space in the offline three-dimensional bin packing problem is quite large, making it difficult for the model to converge as the number of boxes increases. Therefore, reducing the number of valid actions is crucial. In previous studies, many researchers have developed heuristic rules to reduce the number of effective actions. However, some of these heuristic rules drastically reduce the number of valid actions, potentially forgoing the optimal solution, while others do not sufficiently reduce the number of effective actions, making it still challenging for the model to converge when there are many boxes. In response to this, we propose a heuristic rule where boxes are placed only at certain specific locations, which we refer to as Key Points, while other locations are masked. This method integrates well with existing deep reinforcement learning models for solving the offline three-dimensional bin packing problem. We not only theoretically demonstrate the efficacy of this heuristic rule but also empirically show that when our method is combined with existing models, it can easily train with four times the number of boxes. The model converges ten times faster than before, and its performance also improves. Interestingly, even without retraining the model, using our method in the testing phase yields better results than the original method. We also compare our method to other heuristic rules. Experimental results show that our approach strikes a balance between convergence speed and performance.

## 1 INTRODUCTION

The bin packing problem is typically described in terms of the geometric composition of large objects and small items: the large object is defined as empty and needs to be filled with small items. The principal concern is improving the layout of items during the packing process to maximize benefits. From an engineering perspective, the objective of the packing process is usually to maximize the use of raw materials. Even minor layout improvements may result in significant material savings, reducing production costs, which is of great significance for large-scale manufacturers.

Research on the bin packing problem mainly originated in the 1960s (Codd, 1960; Gilmore & Gomory, 1961), initially proposed in relation to cutting stock. Cutting and packing are closely related and do not have a clear boundary. Dyckhoff et al. (1997) emphasized the strong correspondence between cutting and packing from the perspective of the duality of raw materials and space. In this sense, the cutting problem can be seen as placing small items to fill the space of a large object, and conversely, the packing problem can be viewed as cutting a large object into small items. Before the packing problem was developed as a research problem, manual layout strategies were often used, but at a high cost. Over the past few decades, more and more researchers have conducted extensive and in-depth studies on packing problems and achieved some good results (Johnson, 1973; Krause et al., 1975; Maruyama et al., 1977; Garey & Johnson, 1981; Wee & Magazine, 1982), providing great assistance in solving practical problems.

However, with the development of the economy and society, the packing problems encountered in real life are becoming more and more complex, and people are also pursuing higher benefits. In the development process of the entire industry and other industries, such as wood production,

steel production, glass production, etc., transportation naturally plays an important role. Therefore, more complex packing problems have emerged and continue to develop, such as two-dimensional and multi-dimensional packing problems based on one-dimensional packing problems. In fact, the biggest difficulty in the research of packing problems is layout, which is related to the shape, size, and other characteristics of the items to be packed. These are all constraints of the packing problem and are difficult to handle in the solution process.

Due to the success of deep learning (DL) in computer vision, natural language processing, and reinforcement learning (RL) in fields such as Go in recent years, people have begun to study the use of deep reinforcement learning (DRL) to solve combinatorial optimization problems. Various models for solving the offline three-dimensional packing problem have been proposed (Hu et al., 2017; Duan et al., 2019; Jiang et al., 2021; Zhang et al., 2021; Li et al., 2022). From the perspective of effects, they are generally better than heuristic algorithms, but without using some heuristic rules, even the most advanced models can only handle up to 120 boxes at most.

One of the important reasons for not being able to handle more boxes is that compared with other combinatorial optimization problems, the number of actions in the three-dimensional packing problem is too large. The actions in the offline three-dimensional packing problem can be divided into three types. The first type is to choose which box, the number of this type of action is the total number of boxes; the second type is to decide the direction of the box, the number of this type of action is 6; the third type is to decide the position of the box, the number of actions of this type is often the bottom area of the box, generally tens of thousands. The product of the three is the total number of actions. As a comparison, the number of actions in the traveling salesman problem is only the number of cities, far less than the number of actions in the packing problem.

As we can see, the main reason for the excessive number of total actions is the excessive number of position actions. Therefore, we start with reducing the number of position actions to reduce the number of total actions. Therefore, we propose a heuristic rule that sets the packing position to certain specific points, which we call Key Points, greatly reducing the number of actions and accelerating convergence.

Our contributions are as follows:

1. We propose a heuristic rule that integrates well with existing deep reinforcement learning models. From an experimental point of view, for the original model that can only train 100 boxes, after combining with our method, even if 400 boxes are trained, the model can converge, and the convergence speed is significantly faster, and the performance is also improved; compared with other heuristic rules, our method also balances convergence speed and effect well.

2. Theoretically, we prove that even after using our method to mask a large number of actions, the optimal solution will not be missed.

## 2 RELATED WORK

### 2.1 CONVENTIONAL ALGORITHMS

Conventional algorithms for solving the three-dimensional bin packing problem can be categorized into three types: exact algorithms, approximation algorithms, and meta-heuristic algorithms. Owing to the complexity of the Three-Dimensional Bin Packing Problem (3D BPP), there are few exact algorithms. Approximation algorithms are generally designed based on certain heuristic rules. These algorithms tend to solve the problem relatively swiftly, but the quality of the solution relies heavily on the effectiveness of the heuristic rules. A good heuristic rule requires the algorithm designer to have a deep understanding of the bin packing problem. In some cases, we can also conduct some theoretical analyses of approximation algorithms, such as time complexity and worst-case performance. Meta-heuristic algorithms, such as simulated annealing, genetic algorithms, and particle swarm optimization, usually take a longer time and cannot guarantee the quality of the solution. Their advantage is that they generally improve the solution as the time increases, and perform well when there are no good heuristic rules. Due to the complexity of the three-dimensional bin packing problem, the latest traditional methods generally combine approximation algorithms and meta-heuristic algorithms.

Due to space limitations, we can only introduce a few works for each type of algorithm. The current most accurate algorithm (Silva et al., 2019) needs several hours to solve a packing problem with only 12 items, which is obviously not efficient enough for practical use. For approximation algorithms, Crainic et al. (2008) introduced the Extreme Point, and Parreño et al. (2008) proposed the Empty Maximal Space. Both these heuristic rules strictly mandate that boxes can only be situated in certain designated positions. Both rules significantly improved packing efficiency at the time, particularly the Empty Maximal Space. Presently, many newly introduced algorithms only select from a few positions stipulated by the Empty Maximal Space when deciding on the box placement. In terms of meta-heuristic algorithms, there are many at present, with attempts at simulated annealing (Fenrich et al., 1989; Zhang et al., 2007), genetic algorithms (Kang et al., 2012; Corcoran III & Wainwright, 1992; Whitley & Starkweather, 1990; Karabulut & İnceoğlu, 2004; Gonçalves & Resende, 2013; Wu et al., 2010; de Andoin et al., 2022), ant colony optimization Silveira et al. (2013), and quantum algorithms (De Andoin et al., 2022; Bozhedarov et al., 2023; V. Romero et al., 2023).

## 2.2 Learning-based Algorithms

A large part of learning-based algorithms is based on PointerNet (PtrNet)(Vinyals et al., 2015), which is a neural network with a specific attention mechanism, similar to Seq2Seq networks (Sutskever et al., 2014). The main difference between Seq2Seq networks and PtrNet is that the number of target classes output at each step in Seq2Seq networks is fixed, but in PtrNet, the size of the output dictionary is variable. Bello et al. (2016) first attempted to combine PtrNet and RL to solve combinatorial optimization problems in solving the Traveling Salesman Problem (TSP). Hu et al. (2017) was the first to attempt to use PtrNet and RL to generate the box order to solve the new 3D BPP, i.e., designing a bin with the smallest surface area that can pack all the boxes, but the orientation and position are obtained through heuristic methods. Duan et al. (2019) was the first work to use DRL to generate the order and direction of boxes, but the position was obtained through conventional methods. Vaswani et al. (2017) proposed the Transformer. Since box data does not have a temporal order like time series and language, Transformers without position embeddings perform better in BPP than PtrNet. Zhao et al. (2021a) used a heightmap to represent the state of the boxes in the bin and added a convolutional neural network to the model. Jiang et al. (2021) was the first to propose a purely learning-based method, where the index, orientation, and position of the box are all determined by the neural network. Zhang et al. (2021) introduced a heuristic rule for position. When determining the two-dimensional coordinates of the position of the box, one-dimensional coordinates are determined by heuristic methods, and the other dimension is determined by learning algorithms. Li et al. (2022) introduced a heuristic rule for box index, that is, no matter how many boxes there are, decisions are only made among a small number of boxes, generally no more than 30. These small number of boxes are randomly selected, so this method can expand to 1000 boxes, but when the number of boxes is 100, it is not better than (Zhang et al., 2021). Zhao et al. (2021a) only considers positions generated by heuristic rules in online 3D BPP. These heuristic rules include Corner Point(Martello et al., 2000), Extreme Point, Empty Maximal Space (Parreño et al., 2008) and Event Point.

## 3 BACKGROUND

### 3.1 DEFINITION OF THE PROBLEM

All rectangular cuboid boxes need to be packed into a rectangular cuboid bin. The length and width of the bin are fixed, but the height is not. $L, W$ represent the length and width of the bin, respectively. We use a coordinate system to describe the positions of the items and the bin. $(0, 0, 0)$ is defined as the coordinates of the left-front-bottom vertex of the bin. $[N]$ denotes $\{1, 2, ..., N\}$, where $[N]$ is the set of box indices and $N$ is the number of boxes. The length, width, and height of the $i$-th box are $l_i, w_i, h_i$ respectively. After all the boxes are packed into the bin, the coordinates of each item are fixed. $(x_i, y_i, z_i)$ represents the coordinates of the right-rear-top vertex of the $i$-th item. The goal is

$$\min H \tag{1}$$

where $H = \max_{i \in [N]} z_i$.

The constraints are:

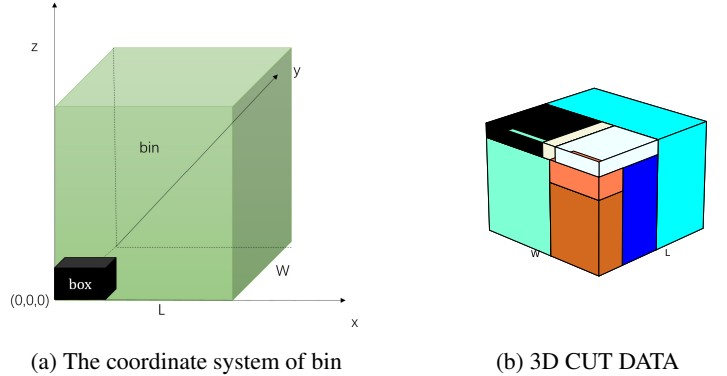

(a) The coordinate system of bin          (b) 3D CUT DATA

Figure 1: The coordinate system of bin and example of 3D CUT DATA

1. Boxes cannot be placed diagonally.
2. Boxes cannot be partially outside the bin.
3. Boxes cannot overlap with each other.

If we are privy to the length, width, and height of all boxes from the beginning, the problem is recognized as offline 3D BPP; if we have knowledge of the dimensions of just one box at each decision stage, suggesting a fixed packing sequence, then the problem falls into the category of online 3D BPP. The primary emphasis of our work lies on offline 3D BPP.

### 3.2 FORMULATING OFFLINE 3D BPP AS A MARKOV DECISION PROCESS

**State** In offline 3D BPP, the state can be divided into two categories: the state of boxes that have not been packed into the bin is generally described by the length, width, and height of these boxes; and the state of boxes that have been packed into the bin. For the state of the boxes that have been packed into the bin, in addition to being described by the length, width, and height, it also involves the interior of the bin. There are currently two main ways to describe the interior of the bin, one is to use a heightmap (Zhao et al., 2021a), which can be regarded as an overhead view of the bin. Using the concept of pixels in the image, each pixel in the heightmap represents height; the other is to directly give the position and orientation of each box in the bin (Li et al., 2022).

**Action** In offline 3D BPP, actions can be divided into three categories. The first is the index action. Given that we select one from $N$ boxes, the number of index actions is $N$; The second is the orientation action. See Figure 2, the number of orientation actions is 6; The third is the position action. The position action can be seen as choosing the x and y coordinates of the left-front-bottom vertex of the box. As to why it does not involve the z coordinate, if the same x, y coordinate corresponds to multiple z coordinates, the smallest z coordinate is taken by default. Therefore, the number of position actions is $L \times W$. Generally, the order of these three types of actions is to determine the index of the box first, then determine the orientation of the box, and finally determine the position of the box. The decision on which action to take is determined by the policy function, which is approximated by a neural network. The output of the neural network is a probability vector, each component of which is non-negative, and the sum of all components is 1. Each component represents the probability of taking the corresponding action.

**Reward** The reward is set as the loading rate $r_u = \frac{\sum_{i=1}^{N} l_i w_i h_i}{LWH}$.

## 4 METHOD

Let's denote $H_{max} = \sum_{i=1}^{N}(l_i + w_i + h_i)$. We represent the set $\{(x, y, z)|a_1 < x < a_2, b_1 < y < b_2, c_1 < z < c_2\}$ using the notation $(a_1, b_1, c_1, a_2, b_2, c_2)$. Define U as the set $\{(x, y, z)|0 \le x \le L, 0 \le y \le W, 0 \le z \le H_{max}\}$. Suppose boxes $j_1, j_2, ..., j_n$ have been placed into the bin, with the coordinates of their left-front-bottom vertices as $(x'_{j_1}, y'_{j_1}, z'_{j_1}), (x'_{j_2}, y'_{j_2}, z'_{j_2}), ..., (x'_{j_n}, y'_{j_n}, z'_{j_n})$.

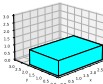 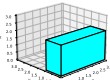 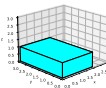 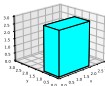 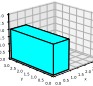 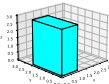

Figure 2: Six orientations.

We define $IS = (\mathbb{R}^3 - U) \cup (\cup_{i=1}^n (x'_{j_i}, y'_{j_i}, z'_{j_i}, x_{j_i}, y_{j_i}, z_{j_i}))$. We represent the empty set as $\phi$.

**Definition 1 (Key Point)** *A point $(x', y', z') \in U$ is deemed a Key Point (KP) if $\forall \delta_1, \delta_2, \delta_3 \geq 0 \wedge \sum_{j=1}^3 \delta_i > 0$, there exists $\delta_4, \delta_5, \delta_6 > 0$, such that $(x', y', z', x' + \delta_4, y' + \delta_5, z' + \delta_6) \cap IS = \phi$ and $(x' - \delta_1, y' - \delta_2, z' - \delta_3, x' - \delta_1 + \delta_4, y' - \delta_2 + \delta_5, z' - \delta_3 + \delta_6) \cap IS \neq \phi$*

Packing the left-front-bottom vertex of a box on a KP is referred to as packing the box on a KP. When no boxes are packed in the bin, there is only one KP, i.e., $(0, 0, 0)$. When a box is packed, the KPs change. Figure 3 provides an illustrative example.

If a face of a box neither contacts other boxes nor the surface of the bin, we refer to this face as a **blank face**. Intuitively, fewer blank faces are desirable. In the extreme case of a 100% loading rate, each of the six faces (up, down, left, right, front, back) of every box either contacts another box or the surface of the bin. We continue to use Figure 3 to illustrate the principle that fewer blank faces yield better results after box placement. Suppose $L, W = 3$, the quantity of blank faces in Figures 3a, 3b, 3c, and 3d are 3, 4, 5, 6, respectively. Upon insertion of three boxes, it can be observed that in Figure 3d, when the second box is packed, $H = 4$; in Figure 3c, after the third box is packed, $H = 4$; in Figure 3b, after four boxes are packed, $H = 4$; and in Figure 3a, after four boxes are packed, $H = 3$. These instances demonstrate that after box placement, fewer blank faces handle larger boxes more easily.

Figure 3a has 3 KPs, while Figures 3b, 3c, and 3d each have 4 KPs. When a box is packed on a KP, the increase in the number of KPs is less than when the box is packed on a non-KP. The fewer KPs, the better, because our method only considers placing boxes at KPs. If there are fewer KPs, it is equivalent to having fewer potential valid positions, which typically means the model can converge faster. We define the set $\Omega_3 = \{(x, y, z) | (x, y, z) \text{ is } KP\}$ to be the set of all KPs.

Next, we describe the decision-making process in offline 3D BPP. We denote the input to the neural network as **X**. The process is as follows:

1. $\pi_s = f_s(\mathbf{X})$: Based on $\pi_s$, we decide on the index of the box $k$.
2. $\pi_o = f_o(\mathbf{X}, l_k, w_k, h_k)$: Based on $\pi_o$, we decide on the orientation of the box $o_k$.
3. $\pi_p = f_p(\mathbf{X}, o_k(l_k), o_k(w_k), o_k(h_k))$: Based on $\pi_p$, we decide on the position of the box.

Where $f_s, f_o, f_p$ are the corresponding neural networks, and $o_k(l_k), o_k(w_k), o_k(h_k)$ represent the measurements of the edges parallel to the $x, y, z$ axes when the box $k$ is placed in orientation $o_k$. Our method only pertains to the modification of $\pi_p$, so we will further discuss the computation process of $\pi_p$. The last layer of $f_p$ is a Softmax layer, and we denote the output of the preceding layer as $\boldsymbol{p} \in \mathbb{R}^{(L \times W)}$. Before feeding into the Softmax layer, $\boldsymbol{p}$ needs to be processed as follows:

$$\boldsymbol{p}[i \times W + j] \leftarrow -\infty, \quad i + o_k(l_k) > L \vee j + o_k(w_k) > W \tag{2}$$

Here, $i \in \{0, 1, ..., L - 1\}, j \in \{0, 1, 2..., W - 1\}$, $\leftarrow$ denotes assignment, and the condition $i + o_k(l_k) > L \vee j + o_k(w_k) > W$ arises because if the $x, y$ coordinates of the box's left-front-bottom vertex are $i, j$, respectively, then a part of the box would inevitably be outside the larger box. At this point, the number of valid position actions is given by $(L - o_k(l_k)) \times (W - o_k(w_k))$. Assuming $L, W = 100$ and the box's length, width, and height are all 60, the number of valid position actions is 3600. Even though the box is large enough, the number of valid position actions remains substantial, necessitating the use of heuristic rules to reduce the number of valid position actions.

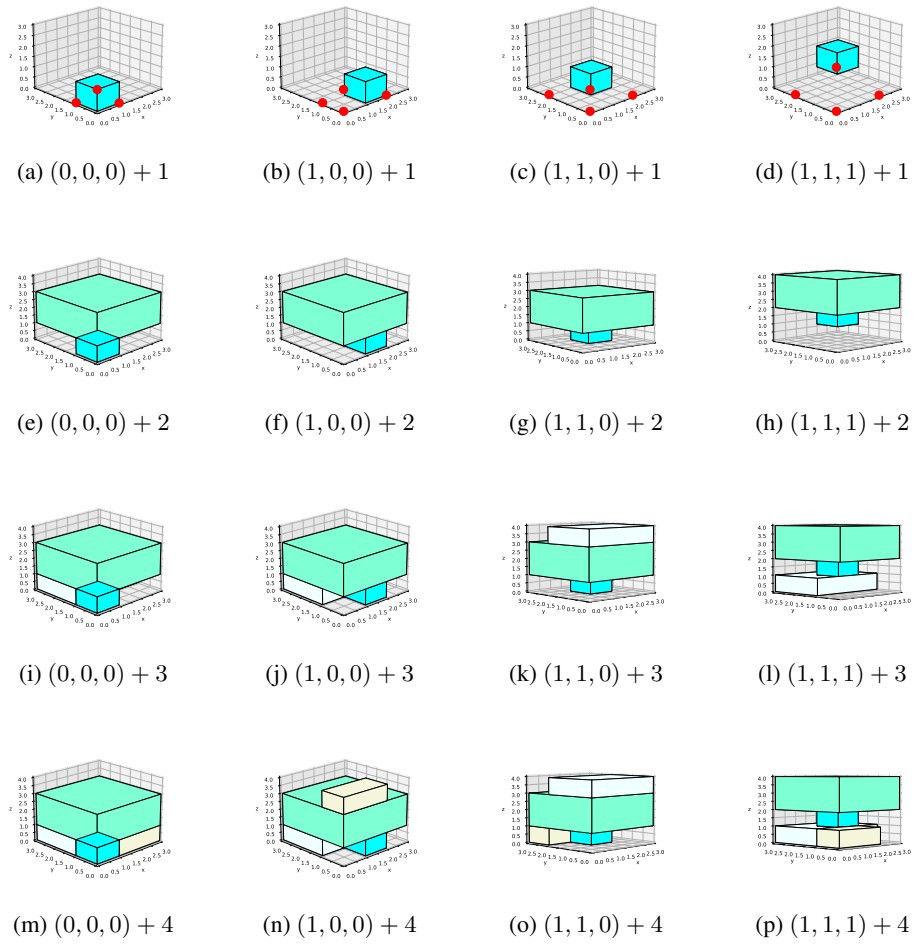

Figure 3: The subtitle $(a, b, c) + d$ denotes that the left-front-bottom vertex of the first box is located at $(a, b, c)$ and that $d$ boxes have already been placed inside. In the first row, a box with dimensions $1 \times 1 \times 1$ is placed in four different positions. The red points indicate the KPs. In Figures 3b and 3c, the highest KP is $(0, 0, 1)$, and in Figure 3d, the highest KP is $(0, 0, 2)$. In the second row, a box with dimensions $3 \times 3 \times 2$ is placed. In the third row, a box with dimensions $3 \times 2 \times 1$ is placed. Finally, in the fourth row, a box with dimensions $2 \times 1 \times 1$ is placed.

Incorporating KP, we propose a novel method. We define condition $C_1$ as $\forall m, if\ (i, j, m) \in \Omega_3, (i, j, m, i + o_k(l_k), j + o_k(w_k), m + o_k(h_k)) \cap IS \neq \phi$ and condition $C_2$ as $\forall m, (i, j, m) \notin \Omega_3$. The modifications made by our method are:

$$\boldsymbol{p}[i \times W + j] \leftarrow -\infty, \quad i, j\ satisfy \quad C_1 \vee C_2 \tag{3}$$

Condition $C_1$ implies that even when the box's left-front-bottom vertex is packed on a KP, it's necessary to ensure that no part of the box is outside the larger box or overlapping with other boxes. Condition $C_2$ implies not considering non-KPs. For both Equation 2 and 3, actions set to $-\infty$ are referred to as invalid actions. Position actions that are not assigned a value of $-\infty$ are termed as valid position actions.

Next, we will demonstrate the effectiveness of our algorithm from a theoretical perspective. We consider a special class of box data, where their optimal solutions precisely correspond to a cuboid with length $L$ and width $W$. We refer to this as 3D CUT DATA. Figure 1b provides an example.

**Theorem 1** *In offline 3D BPP, for 3D CUT DATA, there exists at least one packing order where every box is packed on a KP, achieving the optimal solution.*

Theorem 1 guarantees the existence of an optimal solution, suggesting that an optimal solution of 3D CUT DATA can still be achieved even if we solely contemplate placing the box at a KP. Compared to non-3D CUT DATA, 3D CUT DATA has fewer optimal solutions since no space is wasted in the optimal solutions of 3D CUT DATA. The choice of box placement is also stricter, thereby, we believe that Theorem 1 can be generalized to non-3D CUT DATA.

## 5 EXPERIMENT

We implemented our approach on two state-of-the-art network architectures, namely attend2pack (Zhang et al., 2021) and RCQL (Li et al., 2022).

For both attend2pack and RCQL, we employed Adam (Kingma & Ba, 2014) as the optimizer, with training guided by the Rollout algorithm (Kool et al., 2019). When $p$ in the Rollout algorithm exceeds 0.95, the learning rate is reduced by 5%. The Rollout algorithm is run for 500 epochs, with each epoch consisting of 10 steps, a batch size of 64, and a significance level of 0.05. Both models use LayerNorm (Ba et al., 2016) for normalization. They incorporate multi-head attention layers or their variants, with 8 heads, each of size 16, and 3 such layers or variants. The feedforward layer is composed of two fully connected layers, with output dimensions of 512 and 128, and activation functions being ReQUr (Yu et al., 2021) and ReQU (Li et al., 2019). The test set includes 16384 samples, with lengths, widths, and heights of the boxes in both the test and training sets randomly generated with equal probability from integers between 10 and 50.

For attend2pack-specific hyperparameters, the initial learning rate is set to $10^{-5}$. The $C$ value from the original paper (Zhang et al., 2021) is 10. The model includes 3 convolutional layers, all with 4 output channels. The first convolutional layer has 2 input channels, while the second and third convolutional layers have 4 input channels.

For RCQL-specific hyperparameters, the initial learning rate is $10^{-4}$, the length of the recurrent FIFO queue is 20, the context size of the packed state is 30, and the context size of the unpacked state is $\min\{N, 100\}$.

Our comparison includes both traditional and learning-based algorithms: 1) Genetic Algorithm with Deepest Bottom Left Heuristic (GA+DBLF) (Wu et al., 2010), where the population size and number of generations are set at 120 and 200 respectively; 2) Extreme Point (EP) (Crainic et al., 2008); 3) Largest Area Fit First (LAFF) (Gürbüz et al., 2009); 4) EB-AFIT packing algorithm (Baltacioglu, 2001); 5) MTSL (Duan et al., 2019); 6) Multimodal (MM) (Jiang et al., 2021); 7) attend2pack (A) (Zhang et al., 2021); 8) RCQL (R) (Li et al., 2022).

Table 1 illustrates the experimental outcomes for varying $N$ with $L = 120, W = 100$. K implies the inclusion of our method solely during test truncation, and KT implies the inclusion of our method during both the training and testing stage. We observe that upon incorporating our method during training and testing of attend2pack, the greatest results are achieved across all $N$. Furthermore,

Table 1: The experimental outcomes for varying $N$ with $L = 120, W = 100$. Both figures outside and inside parentheses exclude the % symbol. The figure outside the parentheses denotes the average loading rate following seven experimental repetitions, while the figure within the parentheses signifies the standard deviation.

| $N$ | 25 | 50 | 100 | 200 | 400 |
|---|---|---|---|---|---|
| GA+DBLF | 60.2(1.7) | 62.4(1.6) | 65.9(1.8) | 65.9(1.4) | 61.8(2.4) |
| EP | 62.6(0) | 61.7(0) | 62.9(0) | 65.9(0) | 60.6(0) |
| LAFF | 62.5(0) | 61.4(0) | 60.7(0) | 63.1(0) | 64.3(0) |
| EB-AFIT | 61.4(0) | 60.7(0) | 63.1(0) | 62.3(0) | 61.2(0) |
| MTSL | 65.8(2.0) | 65.7(2.8) | 51.2(2.4) | 52.9(3.4) | 54.4(2.5) |
| MM | 68.0(2.1) | 68.3(2.6) | 68.7(1.6) | 52.0(1.0) | 55.9(1.9) |
| A | 73.0(1.4) | 74.3(1.6) | 73.4(3.5) | 55.7(2.2) | 50.1(1.5) |
| A+K(Our) | 78.0(2.6) | 79.5(1.2) | 78.4(1.4) | 67.9(3.8) | 66.0(3.5) |
| A+KT(Our) | **81.9**(1.8) | **81.8**(0.8) | **82.6**(0.4) | **82.0**(0.6) | **81.8**(0.4) |
| R | 69.8(2.6) | 70.4(1.8) | 71.1(3.8) | 70.5(3.0) | 71.1(1.0) |
| R+K(Our) | 73.3(2.9) | 73.9(1.7) | 73.6(2.2) | 72.7(1.2) | 73.7(1.3) |
| R+KT(Our) | 75.4(1.1) | 75.5(1.2) | 76.7(1.3) | 75.6(1.6) | 75.8(1.4) |

we note that an improvement is seen for either attend2pack or RCQL when our method is incorporated during the testing phase, which is particularly noticeable when the model is attend2pack and $N$ is fairly large. This suggests that improvements can be made without retraining the model. Simultaneously, we observe that after incorporating our method during both training and testing stages, not only does the standard deviation decrease, indicating greater training stability, but also for attend2pack, the model can still converge even when $N$ is larger.

Table 2: The experimental outcomes for varying $L, W$ with $N = 100$. Both figures outside and inside parentheses exclude the % symbol. The figure outside the parentheses denotes the average loading rate following seven experimental repetitions, while the figure within the parentheses signifies the standard deviation.

| (L,W) | (140,120) | (160,140) | (180,160) |
|---|---|---|---|
| GA+DBLF | 60.1(2.0) | 63.3(3.8) | 61.7(2.3) |
| EP | 62.9(0) | 65.2(0) | 64.2(0) |
| LAFF | 61.4(0) | 62.5(0) | 62.2(0) |
| EB-AFIT | 62.5(0) | 60.4(0) | 62.7(0) |
| MTSL | 60.0(3.9) | 61.1(2.3) | 55.2(1.8) |
| MM | 70.7(2.9) | 69.1(3.2) | 70.3(3.4) |
| A | 73.9(2.0) | 74.9(2.6) | 74.8(2.3) |
| A+K(Our) | 78.3(1.4) | 77.3(1.2) | 77.4(2.8) |
| A+KT(Our) | **81.0**(1.3) | **81.9**(0.1) | **81.4**(1.8) |
| R | 72.6(1.7) | 71.7(2.9) | 72.7(2.3) |
| R+K(Our) | 73.9(1.8) | 73.8(2.5) | 73.4(3.4) |
| R+KT(Our) | 75.9(1.0) | 77.0(1.3) | 76.9(2.2) |

Table 2 exhibits the experimental outcomes for varying $L, W$ with $N = 100$. We continue to observe that combining attend2pack with our method surpasses existing models.

We next compare with the four heuristic rules in (Zhao et al., 2021b), namely Corner Point (CP), Extreme Point (EP), Empty Maximal Space (EMS), and Event Point (EV).Table 3 provides the average count of valid position actions (ACOVPA) and initial loading rate (ILR) for the original

model, other heuristic rules, and our approach. From Table 3 and Figure 4, we infer that: the lesser the average count of valid position actions, the quicker the convergence; employing heuristic rules often yields a comparably high initial loading rate, a crucial cause for the hastened convergence post heuristic rule addition; KP's final effect is akin to EMS and EV, but its convergence is significantly faster, partly due to fewer average count of valid position actions and partly due to KP's higher initial loading rate, further indicating that in offline 3D BPP, EMS and EV incorporate numerous superfluous valid position actions; in offline 3D BPP, the performance of EV approximates that of EMS, while the experimental results of online 3D BPP from (Zhao et al., 2021b) indicate that EV surpasses EMS. It has been traditionally believed that offline 3D BPP is more challenging than online 3D BPP due to the action count in offline 3D BPP being $N \times 6 \times L \times W$, while in online 3D BPP, owing to a fixed packing sequence, the action count is $6 \times L \times W$, implying that a smaller action space enables easier learning of the optimal solution.However, as it currently stands, position selection in offline 3D BPP is more flexible. For some heuristic rules that excel in online 3D BPP with regard to position selection, before their application to offline 3D BPP, we should contemplate whether we can develop new heuristic rules building on these. The newly proposed heuristic rules should aim to further curtail the number of valid position actions and demonstrate superior performance in offline 3D BPP.

Table 3: The average count of valid position actions for $N = 100, L = 120, W = 100$.

|  | origin | CP | EP | EMS | EV | KP(Our) |
|---|---|---|---|---|---|---|
| ACOVPA | 6941.4 | 10.0 | 13.2 | 58.9 | 153.8 | 16.7 |
| ILR | 25.2(4.2) | 52.7(1.5) | 53.2(2.6) | 53.2(3.7) | 52.4(3.9) | 54.9(3.4) |

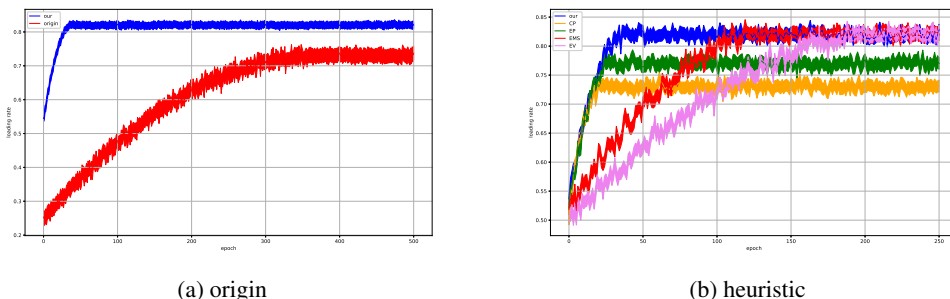

(a) origin           (b) heuristic

Figure 4: The training curve of attend2pack for $L = 120, W = 100, N = 100$. Figure 4a contrasts the training curves of our method with the original model, while Figure 4b contrasts our method with the other four heuristic rules.

## 6  CONCLUSION

In this study, for offline 3D BPP, we introduce KP, a heuristic rule that seamlessly integrates with DRL. Incorporating KP into the initial model enhances both the model's convergence speed and effectiveness. In comparison to other heuristic rules, KP strikes a balance between rate of convergence and performance. Furthermore, our experimental outcomes highlight the differences in employing DRL for solving offline 3D BPP versus online 3D BPP, with the former offering more flexible position selection. Future research will delve into whether it's possible to further decrease the count of valid position actions based on KP, with the aim to expedite model convergence and achieve further enhancement in performance.

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

# A  APPENDIX

## A.1  PROOF OF THEOREM 1

---

**Algorithm 1** packing order where every box is packed on a KP to achieve the optimal solution

---

**Input:** left-front-bottom vertex coordinate and orientation of item set $I = \{(x'_1, y'_1, z'_1, o'_1), (x'_2, y'_2, z'_2, o'_2), ..., (x'_N, y'_N, z'_N, o'_N)\}$
  1: Initialize the KP set $\Omega_3 = \{(0, 0, 0)\}$
  2: **while** $\Omega_3 \neq \phi$ **do**
  3:   **for** $(x', y', z')$ in $\Omega_3$ **do**
  4:     **for** $(x'_i, y'_i, z'_i, o'_i)$ in $I$ **do**
  5:       **if** $x' = x'_i$ and $y' = y'_i$ and $z' = z'_i$ **then**
  6:         $i_{th}$ box is packed on $(x', y', z')$ in orientation $o'_i$. $I \leftarrow I \backslash (x'_i, y'_i, z'_i, o'_i)$, Update $\Omega_3$.
  7:         **break**
  8:       **end if**
  9:     **end for**
 10:   **end for**
 11: **end while**

---

**Proof** *The measure appearing in the proof is assumed to be the Lebesgue measure on $\mathbb{R}^3$. We define the optimal solution of 3D CUT DATA, corresponding to $H$, as $H_{min}$. Subsequently, $U$ is redefined as $\{(x, y, z) | 0 \leq x \leq L, 0 \leq y \leq W, 0 \leq z \leq H_{min}\}$, from which it is apparent that:*

$$\sum_{i=1}^{N} l_i w_i h_i = LW H_{min} \tag{4}$$

*We hypothesize that when achieving the optimal solution, the coordinate of the left-front-bottom vertex of the $i$-th box is $(x'_i, y'_i, z'_i)$, with an orientation of $o'_i$. We term $(x'_i, y'_i, z'_i)$ as the pre-determined position of the $i$-th box. Algorithm 1 provides a boxing order that can reach the optimum and positions the box on the KP at every step. We will prove the effectiveness of Algorithm 1 using proof by contradiction.*

*Clearly, each step of Algorithm 1 positions a box on the KP. If Algorithm 1 cannot achieve the order in Theorem 1, only the following three scenarios could occur:*

  *1. $\Omega_3 \neq \phi$ and $I = \phi$, which corresponds to the existence of KP in $U$, but no boxes remaining.*

2. $\Omega_3 = \phi$ and $I \neq \phi$, which corresponds to the absence of KP in $U$, but surplus boxes still exist.

3. $\Omega_3 \neq \phi$ and $I \neq \phi$, which corresponds to the existence of KP in $U$, as well as boxes that have not been packed into the bin, but none of the remaining boxes' pre-determined positions are KPs.

*We will prove that none of these scenarios will occur.*

*Considering the first scenario, where $\Omega_3 \neq \phi$ and $I = \phi$, corresponding to the existence of KP in $U$, but no boxes remain. Recalling the definition of KP $(x', y', z')$, there exist $\delta_4, \delta_5, \delta_6 > 0$ such that $(x', y', z', x' + \delta_4, y' + \delta_5, z' + \delta_6) \cap IS = \phi$, where $IS = (\mathbb{R}^3 - U) \cup (\cup_{i=1}^N (x'_{j_i}, y'_{j_i}, z'_{j_i}, x_{j_i}, y_{j_i}, z_{j_i}))$. We note the current $\delta_4, \delta_5, \delta_6$. We define $ISS = \cup_{i=1}^N (x'_{j_i}, y'_{j_i}, z'_{j_i}, x_{j_i}, y_{j_i}, z_{j_i})$, clearly, $(x', y', z', x' + \delta_4, y' + \delta_5, z' + \delta_6) \cap ISS = \phi$ We use $m(\Omega)$ to denote the measure of set $\Omega$, because $(x', y', z', x' + \delta_4, y' + \delta_5, z' + \delta_6) \cup ISS \subset U$, we have $m((x', y', z', x' + \delta_4, y' + \delta_5, z' + \delta_6)) + m(ISS) \leq m(U)$, which leads to*

$$\delta_4 \delta_5 \delta_6 + \sum_{i=1}^N l_i w_i h_i \leq LWH_{min} \qquad (5)$$

*Equation 5 and Equation 4 are in conflict.*

*Considering the second scenario, $\Omega_3 = \phi$ and $I \neq \phi$, corresponding to the absence of KP in $U$, but there are surplus boxes. Assume that $n$ boxes have been packed at this point, with $ISS = \cup_{i=1}^n (x'_{j_i}, y'_{j_i}, z'_{j_i}, x_{j_i}, y_{j_i}, z_{j_i})$. We define $S = U - ISS$, which is clearly a closed set. If there exists a KP, it must belong to $S$, with the following equation:*

$$m(S) = m(U) - m(ISS) = LWH_{min} - \sum_{i=1}^n l_i w_i h_i > 0 \qquad (6)$$

*Let's discuss two sub-scenarios.*

*First sub-scenario: $\exists (x', y', z') \in S, \exists \delta_4, \delta_5, \delta_6 > 0, (x', y', z', x' + \delta_4, y' + \delta_5, z' + \delta_6) \cap IS = \phi$. Since $(x', y', z')$ is not a KP, there must exist $\delta_1^{(1)}, \delta_2^{(1)}, \delta_3^{(1)} \geq 0 \wedge \sum_{i=1}^3 \delta_i^{(1)} > 0$ such that $(x' - \delta_1^{(1)}, y' - \delta_2^{(1)}, z' - \delta_3^{(1)}, x' - \delta_1^{(1)} + \delta_4, y' - \delta_2^{(1)} + \delta_5, z' - \delta_3^{(1)} + \delta_6) \cap IS = \phi$. Clearly, $(x' - \delta_1^{(1)}, y' - \delta_2^{(1)}, z' - \delta_3^{(1)}) \in S$ is not a KP. Therefore, we can find $\delta_1^{(2)}, \delta_2^{(2)}, \delta_3^{(2)} \geq 0 \wedge \sum_{i=1}^3 \delta_i^{(2)} > 0$ such that $(x' - \sum_{i=1}^2 \delta_1^{(i)}, y' - \sum_{i=1}^2 \delta_2^{(i)}, z' - \sum_{i=1}^2 \delta_3^{(i)}) \in S$ is not a KP. We can continue this process, defining the sequence of points $a_j = (x' - \sum_{i=1}^j \delta_1^{(i)}, y' - \sum_{i=1}^j \delta_2^{(i)}, z' - \sum_{i=1}^j \delta_3^{(i)})$. Obviously, $\lim_{j \to +\infty} a_j$ exists, so let $a = \lim_{j \to +\infty} a_j$. As $S$ is a closed set, $a \in S$. According to the original assumption, $a$ cannot be a KP. Let $a = (x'_a, y'_a, z'_a)$. By the contrapositive of the definition of KP, $\exists \delta_1, \delta_2, \delta_3 \geq 0 \wedge \sum_{i=1}^3 \delta i > 0$, such that for all $\delta_4, \delta_5, \delta_6 > 0$, when $(x'_a, y'_a, z'_a, x'_a + \delta_4, y'_a + \delta_5, z'_a + \delta_6) \cap IS = \phi$, we have $(x'_a - \delta_1, y'_a - \delta_2, z'_a - \delta_3, x'_a - \delta_1 + \delta_4, y'_a - \delta_2 + \delta_5, z'_a - \delta_3 + \delta_6) \cap IS = \phi$. Clearly, $(x'_a - \delta_1, y'_a - \delta_2, z'a - \delta_3) \in S$, so $\lim_{j \to +\infty} a_j \neq a$, which contradicts the previous condition.*

*Second sub-scenario: $\forall (x', y', z') \in S$ and $\forall \delta_4, \delta_5, \delta_6 > 0, (x', y', z', x' + \delta_4, y' + \delta_5, z' + \delta_6) \cap IS \neq \phi$. We define $int(S)$ as the interior set of $S$, $cl(S)$ as the closure of set $S$, and $\partial S$ as the boundary of set $S$. Since $S$ is a closed set, $S = cl(S)$, and because $cl(S) = int(S) \cup \partial S$ and $int(S) \cap \partial S = \phi$, we have*

$$m(S) = m(int(S)) + m(\partial S) \qquad (7)$$

*Obviously, $\partial S \subset \partial U \cup (\cup_{i=1}^n \partial(x'_{j_i}, y'_{j_i}, z'_{j_i}, x_{j_i}, y_{j_i}, z_{j_i}))$, so we have*

$$m(\partial S) \leq m(\partial U) + \sum_{i=1}^n m(\partial(x'_{j_i}, y'_{j_i}, z'_{j_i}, x_{j_i}, y_{j_i}, z_{j_i})) = 0 \qquad (8)$$

*Therefore, $m(\partial S) = 0$, and from Equations 6 and 7, $m(int(S)) > 0$. For $(x', y', z') \in int(S)$, there exists $\delta > 0$ such that $\{(x, y, z)|(x - x')^2 + (y - y')^2 + (z - z')^2 < \delta\} \subset S$. Hence, we let $\delta' = \frac{\delta}{10}$ and find that $(x', y', z', x' + \delta', y' + \delta', z' + \delta') \cap IS = \phi$, which contradicts the premise.*

*In the third scenario, $\Omega_3 \neq \phi$ and $I \neq \phi$, corresponding to the existence of KP in $U$ and remaining boxes that haven't been packed into the bin, but the pre-determined positions of the remaining boxes are not KPs. We define:*

$$\Omega_z = \{(x, y, z)|(x, y, z) \in \Omega_3 \wedge z = \min_{(x'', y'', z'') \in \Omega_3} z''\} \tag{9}$$

$$\Omega_y = \{(x, y, z)|(x, y, z) \in \Omega_z \wedge y = \min_{(x'', y'', z'') \in \Omega_z} y''\} \tag{10}$$

$$\Omega_x = \{(x, y, z)|(x, y, z) \in \Omega_y \wedge x = \min_{(x'', y'', z'') \in \Omega_y} x''\} \tag{11}$$

*It is apparent that $\Omega_x$ has only one element, which we designate as point $b = (x_b, y_b, z_b)$. Next, we prove that for KP $b$, if the box is not packed on $b$, then $b$ remains a KP. We prove this in two steps:*

1. *For all $\delta_1, \delta_2, \delta_3 \geq 0 \wedge \sum_{i=1}^{3} \delta_i > 0$, and for all $\delta_4, \delta_5, \delta_6 > 0$, we have $(x_b - \delta_1, y_b - \delta_2, z_b - \delta_3, x_b - \delta_1 + \delta_4, y_b - \delta_2 + \delta_5, z_b - \delta_3 + \delta_6) \cap IS \neq \phi$.*

2. *After a new box is packed and it's not packed on $b$, the condition $\exists \delta_4, \delta_5, \delta_6 > 0, (x_b, y_b, z_b, x_b + \delta_4, y_b + \delta_5, z_b + \delta_6) \cap IS = \phi$ still holds.*

*For the first step, suppose there exist $\delta_1^{(1)}, \delta_2^{(1)}, \delta_3^{(1)} \geq 0 \wedge \sum_{i=1}^{3} \delta_i^{(1)} > 0$, and $\delta_4, \delta_5, \delta_6 > 0$ such that $(x_b - \delta_1^{(1)}, y_b - \delta_2^{(1)}, z_b - \delta_3^{(1)}, x_b - \delta_1^{(1)} + \delta_4, y_b - \delta_2^{(1)} + \delta_5, z_b - \delta_3^{(1)} + \delta_6) \cap IS = \phi$. This is similar to the first sub-scenario of the second scenario. Since $(x_b - \delta_1^{(1)}, y_b - \delta_2^{(1)}, z_b - \delta_3^{(1)})$ is not a KP, we can construct a sequence of points $b_j = (x_b - \sum_{i=1}^{j} \delta_1^{(i)}, y_b - \sum_{i=1}^{j} \delta_2^{(i)}, z_b - \sum_{i=1}^{j} \delta_3^{(i)})$. Clearly $\lim_{j \to +\infty} b_j$ exists, so let $B = \lim_{j \to +\infty} b_j$ and $B = (x_B, y_B, z_B)$. Obviously, $x_B \leq x_b, y_B \leq y_b, z_B \leq z_b$ and $x_B + y_B + z_B < x_b + y_b + z_b$. However, according to Equations 9, 10, and 11, $\forall (x, y, z) \in \Omega_3 - \Omega_x$, it is not true that $\neg \exists x \leq x_b \wedge y \leq y_b \wedge z \leq z_b$. This is a contradiction.*

*In the second step, assume a new box is packed but not packed on $b$. For all $\delta_4, \delta_5, \delta_6 > 0$, we have $(x_b, y_b, z_b, x_b + \delta_4, y_b + \delta_5, z_b + \delta_6) \cap IS \neq \phi$. Let the coordinate of left-front-bottom vertex of the new box be $(x'_k, y'_k, z'_k)$ and the top right corner coordinates be $(x_k, y_k, z_k)$. It is clear that $(x_b, y_b, z_b) \in (x'_k, y'_k, z'_k, x_k, y_k, z_k)$. However, based on the conclusion of the first step, $\forall \delta_1, \delta_2, \delta_3 \geq 0 \wedge \sum i = 1^3 \delta_i > 0, \forall \delta_4, \delta_5, \delta_6 > 0$, when $(x_b - \delta_1, y_b - \delta_2, z_b - \delta_3, x_b - \delta_1 + \delta_4, y_b - \delta_2 + \delta_5, z_b - \delta_3 + \delta_6) \cap IS = \phi$, we have $(x'_k, y'_k, z'_k, x_k, y_k, z_k) \cap IS \neq \phi$. This implies that the boxes overlap or a part of the box is placed outside the bin, contradicting the constraints.*

*With the completion of the proofs in the first and second steps, we have proven that for KP $b$, if the box is not packed on $b$, then $b$ remains a KP. According to this conclusion, even after the bin is filled with boxes, a KP still exists. This is identical to the first scenario, which we have already disproven.*□

