# OpenReview forum: "Key point is key in resolving the offline three-dimensional bin packing problem"
_ICLR.cc/2024/Conference — ICLR 2024 Conference Withdrawn Submission_

### Official Review · Reviewer_TnxE · 2023-10-29

**Soundness:** 2 fair
**Presentation:** 1 poor
**Contribution:** 3 good
**Rating:** 3
**Confidence:** 4

**Summary:**

The paper proposes to use the key point as the candidate box placement location to reduce the action space of BPP.  They also prove that "there exists at least one packing order where every box is packed on a KP, achieving the optimal solution." The results show the proposed method could train a model with up to 400 boxes and improve performance and convergence speed. However, the paper is not easy to follow. The key concept such Key Point is not well explained. The details about the method and data configuration are missing.

**Strengths:**

The proposed Key Point seems work well with the RL method.

The authors also present a proof.

The code is provided, which improves the reproducibility.

**Weaknesses:**

The paper is not easy to follow.  The key contribution of the paper is Key Point. However, the key concept is not well explained, which makes the paper confusing. The main ideas and method behind the Key Point are not clear:
1) How to identify and update the Key Point？
2) What's the geometrical meaning of the Key Point?
3) What's the main difference between Key Point with traditional methods such as Corner Point?
4) The proof is a bit hard to follow. It would be better to describe the general ideas and steps before the proof.

Data configurations are not presented.

**Questions:**

How to identify and update the Key Point？

What's the geometrical meaning of the Key Point?

What's the main difference between Key Point with traditional methods such as Corner Point?  What are the key ideas behind it?

---

### Official Review · Reviewer_RRYs · 2023-10-30

**Soundness:** 3 good
**Presentation:** 3 good
**Contribution:** 1 poor
**Rating:** 1
**Confidence:** 5

**Summary:**

The paper solves a version of the three dimensional bin packing problem (BPP) using deep reinforcement learning. The paper identifies the huge action space of the of the BPP as being a critical roadblock to finding good solutions and proposes a mechanism to reduce the size of the action space, namely the identification of "key points". The authors define key points and prove that there exists a packing order of the boxes in which only uses key points will lead to the optimal solution.

**Strengths:**

1. The paper is relatively clear.

Unfortunately, I do not have anything else positive to say about this work in the context of ICLR. If this was an OR journal, it would be an entirely different review.

**Weaknesses:**

There is nothing new in this paper for the ICLR community. The learning mechanisms are known, the problem is known. The only thing that is new is the identification of the "key points". However, I am not aware of any recent papers in the operations research (OR) community that does not use "key points" to solve packing problems. It is entirely obvious that these points must be used. I am not an expert on container packing, so I admit that I do not know if there is a proof out there that these points will lead to an optimal solution. However, I know enough about the problem to confidently say this paper has no place at ICLR. Accepting this paper would constitute running around the OR reviewers who are experts at this domain and can definitely say whether this is an interesting proof or not. The proof has literally no bearing on learning: the "key points" are essential to any heuristic (or exact) technique. I encourage the authors to submit to a journal like EJOR or C&OR where packing problems have been a focus of many works.

I am also not sure about the application in this paper being all that important or relevant. The bin packing problem typically seeks to minimize the total number of bins used, the metaphor being packing trailers for trucks, or containers for airplanes or ships. Minimizing the height (or any dimension, really, since it is irrelevant which one is minimized) is not a problem type I am familiar with.

Another note on the experiments: these just use a bunch of randomly generated instances. Multiple benchmarks for bin packing exist (e.g., Bischoff and Ratcliff (1995) and Davies and Bischoff (1999) just to name two). Why not use these? Real problems generally have some structure to them and are not just a bunch of random things thrown together in a container.

Finally, I would like to point out the tautological title that can only be considered clever if we ignore the fact that the authors literally call the points "key points" and then say they are key... because they are key points.

**Questions:**

I have no questions for the authors, there is nothing that will change my opinion.

---

### Official Review · Reviewer_z1aJ · 2023-10-31

**Soundness:** 2 fair
**Presentation:** 1 poor
**Contribution:** 2 fair
**Rating:** 3
**Confidence:** 5

**Summary:**

This article primarily discusses the introduction of a heuristic rule called KP into offline 3D bin packing problems and its integration with Deep Reinforcement Learning (DRL). The rule aims to reduce the number of effective actions to expedite model convergence and enhance performance.

**Strengths:**

1. This paper introduces a novel heuristic rule that can assist in reducing the action space in 3D bin packing problems.
2. The authors' method demonstrated the best performance in the experiments they designed.

**Weaknesses:**

1. The concept of KP proposed by the authors and their attempt to provide some proofs based on this concept are presented in a rough and overly complex manner.
For instance, IS = (R3 − U), where R3 is not defined in the paper, and the physical meaning of IS should also be elucidated.
In Definition 1, the key point is symbolically defined, yet there is a lack of detailed illustration, insight, or algorithm design for computing the key point. It's difficult for readers to grasp the concept of the key point and determine if it's a reasonable design—for instance, whether it might lead to infeasible solutions or if the computational complexity of the KP algorithm itself is too high.
In the appendix, the authors provide proofs related to KP, but the overly symbolic expressions appear confusing and tiresome. Perhaps outlining their insights might alleviate the readers' burden.
The authors should reorganize their presentation to make their core contribution precise and succinct. If they enhance the clarity of their article, I would consider raising the score.

2. The authors' contribution seems overly emphasized, given that offline packing methods (such as attend2pack, RCQL) and candidate-based packing methods (Zhao et al. (2021a)) have already been introduced. Moreover, I noticed that the authors' method largely draws from these existing works during the implementation process. The authors' contribution solely introduces a new heuristic rule, KP, and I believe these limited contributions might struggle to meet the standard for acceptance at ICLR.

3. Restricting the action space is a sound approach, as verified in existing works (Zhao et al. (2021a)) and observed in our own practice. I'm surprised to see that KP outperforms action designs like EMS and EV with an average of only 16.7 valid action counts. I'm also curious why EP and EMS perform equally, as in previous research, EMS had significantly outperformed EP. While examining the authors' code, I noticed they only submitted their own work's code and did not provide the baseline code. I believe submitting all baseline code for verification is necessary to assess the correctness of the authors' conclusions.

**Questions:**

What does neural network input X represent? I didn't seem to find a direct correspondence in this paper; does this represent all observable objects in the offline setting?